# Effect of Polybutylene Succinate Additive in Polylactic Acid Blend Fibers via a Melt-Blown Process

**DOI:** 10.3390/molecules28207215

**Published:** 2023-10-22

**Authors:** Benchamaporn Tangnorawich, Areerut Magmee, Nanjaporn Roungpaisan, Surachet Toommee, Yardnapar Parcharoen, Chiravoot Pechyen

**Affiliations:** 1Department of Physics, Faculty of Science and Technology, Thammasat University, Khlong Luang 12120, Thailand; seebungkert@gmail.com; 2Thammasat University Center of Excellence in Modern Technology and Advanced Manufacturing for Medical Innovation, Thammasat University, Khlong Luang 12120, Thailand; 3Department of Materials and Textile Technology, Faculty of Science and Technology, Thammasat University, Khlong Luang 12120, Thailand; 4Department of Textile Chemistry Engineering, Faculty of Engineering, Rajamangala University, Khlong Luang 12120, Thailand; 5Industrial Arts Program, Faculty of Industrial Technology, Kamphaeng Phet Rajabhat University, Mueang 62000, Thailand; 6Chulabhorn International College of Medicine, Thammasat University, Khlong Luang 12120, Thailand

**Keywords:** polylactic acid (PLA), polybutylene succinate (PBS), PLA/PBS composite, melt-blown, nonwoven, surface morphology, mechanical strength, thermal properties, crystallinity, chemical groups

## Abstract

This work aimed to study the influence of the polybutylene succinate (PBS) content on the physical, thermal, mechanical, and chemical properties of the obtained polylactic acid (PLA)/PBS composite fibers. PLA/PBS blend fibers were prepared by a simple melt-blown process capable of yielding nanofibers. Morphological analysis revealed that the fiber size was irregular and discontinuous in length. Including PBS affected the fiber size distribution, and the fibers had a smoother surface with increased amounts of added PBS. Differential scanning calorimetry analysis (DSC) revealed that the crystallization temperature of the PLA sheet (105.8 °C) was decreased with increasing PBS addition levels down to 91.7 °C at 10 wt.% PBS. This suggests that the addition of PBS may affect PLA crystallization, which is consistent with the X-ray diffraction analysis that revealed that the crystallinity of PLA (19.2%) was increased with increasing PBS addition up to 28.1% at 10 wt% PBS. Moreover, adding PBS increased the tensile properties while the % elongation at break was significantly decreased.

## 1. Introduction

Green development has emerged as the widely accepted approach to addressing environmental issues stemming from white pollution. The increasing exploration and production of biodegradable materials signify a progressive shift, with these materials gradually finding applications in various industries, including packaging, clothing, healthcare, and others [1]. Recent advancements in biodegradable and biocompatible synthetic polymers, as highlighted by Xu et al. [2], are enhancing their utility in healing skin wounds. Consequently, the design of biodegradable wound dressings holds promise for reducing environmental impact and potentially benefiting patient healing.

Recent developments in nonwoven fabric technology have centered on creating biodegradable and environmentally friendly materials. Nonwoven melt-blown processes have traditionally yielded conventional wound dressings crafted from synthetic materials like polypropylene. Nevertheless, progress may have occurred in leveraging nonwoven melt-blown technology to produce biodegradable wound dressings [3]. The nonwoven process is a direct tailoring method that circumvents the need for a weaving machine and relies on thermal or chemical mechanisms. This process boasts the advantage of tailoring fibers with diverse characteristics and properties tailored to different applications [4]. Various nonwoven techniques have been developed, including electrospinning, forcespinning, meltspinning, spun bond, and melt-blown [5,6,7,8,9]. Among these, the melt-blown process has garnered substantial attention from researchers due to its capacity to produce submicron-diameter fibers. Furthermore, this process is characterized by simplicity, cost-effectiveness, and versatility in various applications.

One of the most popular materials is polylactic acid (PLA), a biodegradable polymer with excellent mechanical properties and biocompatibility. Nevertheless, its flexural strength and low crystallization rate present significant hurdles to its widespread application [10]. The strength of PLA can be improved in various ways, such as by adding inorganic nanoparticles or mixing with other polymers [11]. The polymer used must be a flexible polymer to toughen the PLA. Highly elastic polymers, such as natural rubber and nitrile butadiene rubber, can significantly improve the toughness of PLA, but this can also destroy the biodegradability of the PLA materials. Instead, biodegradable materials, such as poly(butylene succinate) (PBS) [12,13], poly(butylene adipate-co-terephthalate) [14], and polycaprolactone [15], are blended with PLA to toughen it and maintain the biodegradability of the PLA [16].

Among the materials mentioned above, polybutylene succinate (PBS) is the most widely employed polymer. This preference stems from its exceptional attributes: high flexibility, impressive impact strength, elevated elongation at break, biodegradability, and a rapid crystallization rate conducive to efficient industrial molding processes. Additionally, PBS is readily accessible and cost-effective, positioning it as a favorable biomaterial [17,18]. Consequently, the integration of PBS into biodegradable PLA materials presents a unique challenge as it has the potential to enhance PLA’s toughness while preserving its biodegradability significantly.

In the past, there have been many studies to improve the properties of PLA through various preparation methods, but these have yielded different results. For instance, preparing PLA/PBS composites through meltspinning has demonstrated high fiber compatibility. Incorporating approximately 12 wt% PBS increased the crystallinity of PLA and led to a 39% reduction in tensile strength and a 40% increase in elongation at break [19]. Similarly, previous investigations into PLA/PBS fibers, fabricated with varying weight-to-weight (*w*/*w*) ratios using electrospinning, revealed that augmenting the PBS content lowered the glass transition temperature (T_g_) while boosting fiber crystallinity. Furthermore, PBS exhibited a plasticizing effect, enhancing the flexibility, elasticity, and thermal resistance of the resultant PLA/PBS composite fibers compared to pure PLA fibers [20]. Adding PBS elevated PLA’s crystallization rate and decreased the cold crystallization temperature (T_cc_) compared to pure PLA. This phenomenon stemmed from PBS nucleation, prompting PLA crystallization during the cooling process [21]. The specific properties and applications of nonwoven materials made from PLA/PBS blends can vary depending on factors such as the blending ratio, processing method, and additional treatments or additives [17].

This study prepared PLA/PBS composites using a melt-blown process, employing varying PLA: PBS (*w*/*w*) ratios. To investigate the impact of PBS addition on the properties of the resulting PLA/PBS composites, we conducted a comprehensive analysis encompassing the examination of morphology, mechanical strength, thermal characteristics, crystallinity, and chemical groups. These analyses were performed using scanning electron microscopy (SEM), an Instron universal testing machine, differential scanning calorimetry (DSC), X-ray diffraction (XRD), and Fourier transform infrared spectroscopy (FT-IR).

## 2. Results and Discussion

### 2.1. Microstructure Characterization

Table 1 presents fabric formation ratios in the melt-blown process, indicating the presence (forming) or absence (non-forming) of fabric-like structures. Consequently, it is advisable to focus exclusively on the forming ratios for further research and investigation.

The morphology of the nonwoven PLA/PBS composite fibers was examined through SEM analysis, revealing that the PLA/PBS fibers were randomly arranged and exhibited varying sizes (Figure 1). A comparison between pure PLA fibers (Figure 1b) and pure PBS fibers (Figure 1e) illustrated that PBS fibers possessed a larger diameter and smoother surface in contrast to PLA fibers. In the PLA/PBS composite with 2.5 wt.% PBS, small fibers with a uniform distribution and high fiber density, akin to the pure PLA sheet, were observed. However, these fibers were more densely clustered. The resulting fibers exhibited minimal defects in the PLA/PBS composite with 5.0 wt% PBS, featuring a smoother surface, although the fiber density was lower than the composite with only 2.5 wt% PBS. Nevertheless, upon increasing the PBS content to 10 wt%, the fibers displayed an enlarged size and reduced fiber density. Consequently, augmenting the PBS content in the PLA/PBS composites had a discernible impact on fiber size, density, and overall performance.

Under all conditions, it is noteworthy that defects may have arisen during the fabrication process, which was executed via high-speed precursor blowing, possibly leading to intermittent feeding of the precursor [22]. Additionally, polymer fragments were identified attached to the fibers, which may have resulted from the high air velocity causing breaks in the polymer streams before they were attenuated into thin fibers [23].

Figure 2 displays SEM images of the surface of the PLA/PBS blend fibers at 5000× magnification. Pure PLA fibers exhibited a relatively rough surface, whereas pure PBS fibers displayed a relatively smooth surface. The introduction of PBS led to smoother fibers as the PBS content increased, with the PBS phase being uniformly distributed throughout all the composite fibers. Consequently, an effective PBS dispersion enhanced the surface quality of the fibers.

### 2.2. Thermal Properties

The thermal properties of the PLA/PBS fibers were examined through DSC analysis, and the findings are presented in Figure 3 and Table 2. Pure PLA exhibited a T_g_ of 60.5 °C, T_cc_ of 105.8 °C, a melting temperature of 166 °C, and a calculated percent crystallinity (X_c_) of 19.2%. The degree of crystallinity is subsequently defined as
%crystallinity (X_c_) = (∆H_f_ − ∆H_c_)/(∆H°_f_) × 100
where ΔH_f_ is the enthalpy of fusion, ΔH_c_ the enthalpy of crystallization, and ∆H°_f_ the heat of fusion of the completely crystalline materials at the equilibrium melting temperature, T_m_^o^. All are measured at different temperatures, and no changes are made for the change in specific heat. Nevertheless, this method has appeared as a recommended method [24].

Pure PBS exhibited a lower T_cc_ (97.8 °C), melting temperature (114.8 °C), and X_c_ (64.8%), but there was no significant difference in its T_g_ (59.2 °C). The addition of PBS to PLA reduced T_cc_, decreasing from 105.8 °C in pure PLA to 91.7 °C for PLA with a 10 wt% PBS addition. This decrease in T_cc_ can be attributed to the formation of PBS, which facilitated the growth of PLA fibers, as evidenced by the increasing X_c_ with rising PBS content. Consequently, incorporating PBS into PLA can enhance the thermal resistance and crystallinity of the blends [20]. It is well-known that polymer crystallization from the molten state involves two stages: first, nucleation, which can be either homogeneous or heterogeneous, and, second, crystal growth. In this study, the presence of PLA and crosslinking points hindered the formation of perfect crystals within the PBS polymer chain, likely the primary reason for the reduction in *X_c_*(*PBS*). However, the presence of PBS and crosslinking points promoted the formation of more crystal nuclei through heterogeneous nucleation, hence serving as the primary reason for the increase in *X_c_*(*PLA*).

### 2.3. Crystal Structure Characterization

The crystal structure was explored through XRD analysis, and representative results are presented in Figure 4. The characteristic X-ray peak of PLA appeared as a broad peak within the range of 16.2–31.2°, indicative of its amorphous structure [18,20]. In contrast, due to its high crystallinity, PBS exhibited clear peaks at approximately 19.4°, 22.3°, and 29.1°, corresponding to the monoclinic structure or alpha phase [25,26,27]. Adding 2.5 and 5.0 wt% PBS to PLA yielded no significant alteration in the XRD pattern compared to pure PLA. However, when 10 wt% PBS was added to PLA, the XRD pattern diverged from that of PLA with a 5 wt% PBS addition, displaying a distinct combination of PLA and PBS with potentially heightened PLA crystallinity. The absence of characteristic PBS peaks when adding PBS to PLA at 2.5 and 5.0 wt.% may suggest that the PBS content did not reach a level sufficient to detect the signal.

When one component initially exists at a low concentration in polymer blends, the dispersed particles often adopt fibrillar shapes, establishing a dispersed phase-matrix morphology. However, as the concentration of the minor phase increases, the particles draw closer, eventually reaching a percolation threshold. Beyond this threshold, a more substantial portion of the minor component integrates into a single percolating structure, resulting in dual-phase continuity or co-continuity [28]. Within this structure, both phases maintain continuous connectivity throughout the bulk of the blend.

### 2.4. Chemical Structure Characterization

The functional chemical groups of the various PLA/PBS fibers were examined through FT-IR analysis, a valuable analytical technique employed in investigating polymer blends and mixtures. FT-IR offers valuable insights into the interactions between PLA and PBS molecules. The results reveal distinct shifts in absorption bands and the emergence of new peaks in the FT-IR spectra, indicating interactions or compatibility between the two polymers. The positions of the peaks, the peak widths at half height, and the percentage integral absorbance of the C=O bands component are akin to those observed in the PLA/PBS blends.

Figure 5 illustrates the vibrational peak of PLA at 2946 cm^−1^ and 1754 cm^−1^, attributed to C-H stretching (symmetric stretching vibration of CH_3_) and C=O stretching (stretching of the ester group), respectively. These peaks gradually diminish as the PBS concentration increases, with only the 2946 cm^−1^ peak remaining [29]. The strong absorption peak at 1754 cm^−1^, associated with the ester group of PBS, slightly increases with elevated PBS concentration [27]. Another peak at approximately 1085 cm^−1^ corresponds to C-O-C stretching [30].

For PBS, the peaks consist of O-H stretching at 3450 cm^−1^, C-H stretching at 2940 cm^−1^, C=O at 1714 cm^−1^, and C-O stretching at 1150 cm^−1^. The FT-IR spectra of the PLA/PBS composite fibers, with a PBS addition of up to 5 wt.%, mirror that of pure PLA. However, in the case of a 10 wt% PBS addition, the characteristic peaks of both PLA and PBS become evident, featuring two carbonyl peaks at 1754 and 1714 cm^−1^, respectively. The presence of these two carbonyl peaks confirms the intermixing of PLA and PBS. In addition to the interactions within the chain backbones, side groups are prominently associated with adjacent heterogeneities. Another peak, attributed to the stretching vibration of –CH in folded PLA chains and the bending vibration of –OH groups of PBS (around 1128 cm^−1^ and 1150 cm^−1^, respectively) exhibits slight variations as the PBS blend concentration increases [31]. These bond shifts result from strengthening bond energy within the backbones and side groups, reflecting robust intermolecular and intramolecular interactions. These interactions include forming hydrogen bonds between the oxygen functional groups carried by PLA and PBS.

### 2.5. Mechanical Properties

The tensile strength of the PLA/PBS sheets is presented in Figure 6a, demonstrating an increase in tensile strength as the PBS content increases. Interestingly, adding 10 wt% PBS boosted the tensile strength to approximately 0.5 MPa, four times that of pure PLA and roughly 80% pure PBS. Conversely, adding PBS decreased the elongation percentage at the breakpoint (Figure 6b). This aligns with the findings of the morphology analysis, where SEM analysis revealed that the incorporation of PBS reduced sheet defects and enhanced sheet strength. Conversely, the percentage of elongation at the breakpoint decreased as the PBS content increased. High crystallinity, as evident in this study, indicates well-ordered molecular chains, resulting in stiffness and brittleness. Consequently, this leads to high tensile strength but low elongation at the break, a common characteristic in materials such as PLA [32].

PBS is a biodegradable thermoplastic with some flexibility. When added in low concentrations to another material, it may not exhibit rubber-like properties, particularly if the other material possesses high crystallinity and stiffness [33]. The elevation in crystallinity levels becomes evident with increased PBS content, as demonstrated in Section 2.2 DSC analysis. It is worth noting that the distance between the injection point and the collector must remain at room temperature to facilitate the cooling of the sheets. Therefore, the setup distance in our study may have required adjustment, potentially resulting in incomplete cooling of the sheets [25].

The fracture mechanisms were assessed by directly observing fractured surfaces after tensile failure. In contrast to the flat and smooth topography observed in pure PLA, the PLA/PBS composites exhibited abundant plastic deformation characterized by nanofibrillar and wormlike extensions along the tensile direction. Of particular interest was the observation of highly aligned nanofibrils with tapered fracture ends, suggesting the dissipation of a substantial amount of energy through profound stress transfer along the nanofibril backbones or networks. Another significant energy-dissipating mechanism contributing to the high toughness was the formation of nanosized voids, which could grow, merge, and reorient within the channel base.

It is worth noting that these slit-like nanovoids exhibited a regioselective distribution near the nanofibrils, possessed nanosized widths as low as tens of nanometers, and displayed elongated streaks that orderly aligned along the deformation direction. These distinctive characteristics enabled the nanovoids to function as unique energy-dispersive catalysts until the complete consumption of the deformable phase, rather than typical cavities leading to catastrophic fracture. The hierarchical silk-mimicking integrity thus achieved a high deformation capability through various robust energy-dissipating mechanisms while simultaneously balancing the necessity of transferring the applied stress to avoid sacrificing elastic resistance, thereby sustaining high strength.

## 3. Experimental

### 3.1. Materials

The PLA (NatureWorks^®^ Ingeo™ 3251D Injection Grade PLA) used in this study is in pellet form and was produced by Natureworks LLC (Minnetonka, MN, USA). It has a melting point (T_m_) of 170 °C, a Glass Transition Temperature (T_g_) of 65.0 °C, and a 1.24 g/cm^3^ density. Its melt flow index (MFI) is 24 g/10 min, designed for injection applications. The PBS (FZ Type and FZ78tm-grade) is sourced from Biopolymer^TM^ (Thailand).

It has a melting point (T_m_) of 115 °C, a 1.31 g/cm^3^ density, and an MFI of 27.69 g/10 min.

### 3.2. Preparation of PLA/PBS Nonwoven Fibers via a Melt-Blown Process

The melt-blown process is a one-step, solvent-free, and high-throughput fabrication technique that converts solid polymer directly into a nonwoven mat (Figure 7). During this process, the polymer material is melted, with the pellets mechanically sheared by an endlessly rotating screw at a high temperature. The extruder comprises three distinct zones: (1) feed, (2) transition, and (3) metering zones. Subsequently, the molten polymer is passed through a die containing several small-diameter holes, transforming the polymer into filaments. These filaments are then stretched using forced hot air onto a high-speed collector to create a nonwoven fabric [34], as illustrated in Figure 7.

For sample preparation via the melt-blown process, PLA and PBS pellets underwent drying in a vacuum oven at 80 °C for 4 h to eliminate residual moisture. The PLA to PBS ratio choice in a blend or composite material can significantly vary based on specific applications and desired properties. This study primarily focuses on applications related to wound dressings, which necessitate particular attributes, such as flexibility, biocompatibility, moisture management, and ease of application. As a result, considering insights from the existing literature [18,23,26,35], the researcher determined the PLA and PBS ranges as follows. Samples were fabricated at different PLA to PBS weight ratios, specifically 100/0, 97.5/2.5, 95/5, 90/10, 80/20, 70/30, 60/40, 50/50, and 0/100, at various temperatures ranging from 180 to 250 °C for 10 s. Subsequently, a melt flow indexer assessed the MFI of the PLA/PBS mixture using a test weight of 2.16 kg to ascertain the optimal molding temperature during the melting process. The samples were then weighed, and the MFI was calculated. PLA/PBS nonwovens were produced using the melt-blown technique, employing a single-screw laboratory extruder (Model SR V-N-28, brand SR-RUDER BAMBI) equipped with a three-hole head. The die diameter was 0.35 mm, and the die temperature was maintained at 275 °C. The hot air temperature reached approximately 260 °C with a drawing air flow pressure of 0.4 MPa, and the collector distance was set at 45 cm. PLA and PBS pellets were introduced into the extruder, where they melted and were then fed into the spinning head through the extruder. The polymer melt was expelled through the dies, resulting in the deposition of nonwoven samples on the rotating drum (collector). Our preliminary experiments guided the determination of optimal processing parameters presented in Table 3.

### 3.3. Characterization

The morphology of the PLA/PBS blend fibers was observed using a Scanning Electron Microscope (SEM), specifically the JEOL JSM-6610LV model, equipped with an Oxford X-Max 50 detector. SEM was operated at an accelerating voltage of 20 kV, offering magnifications of 250× and 5000×. Before imaging, the samples underwent a gold coating process under vacuum conditions.

To evaluate the mechanical properties of the PLA/PBS nonwoven sheet, rectangular sheets measuring 1 × 10 cm^2^ were cut, with a thickness of approximately 0.50 mm. Tensile testing was conducted by the ASTM-D638 standard [36], employing an Instron Universal Testing Machine (INSTRON 5560) and a Texture Analyzer. The tests were performed at a crosshead speed of 20 mm/min, with a minimum of five samples tested for each pure or composite sheet.

The thermal properties of the samples were analyzed through Differential Scanning Calorimetry (DSC) using a DSC 214 instrument from NETZSCH under a nitrogen (N_2_) atmosphere at a flow rate of 20 mL/min. The heating process involved ramping the sample from 25 to 200 °C at a rate of 10 °C/min, followed by an isothermal hold at 200 °C for 5 min to eliminate thermal history. Subsequently, the sample was cooled to 25 °C at the same rate under an N_2_ atmosphere and then reheated to 200 °C using the same heating rate [37].

The crystal structures of the PLA/PBS nonwoven materials were examined through X-ray Diffraction (XRD) analysis using a D8 Advance instrument from Bruker AXS (Thailand). Ni-filtered Cu Kα radiation (λ = 1.54060 Å) was generated at a voltage of 30 kV and a current of 10 mA, with a scan speed of 5°/min. XRD data were collected within a 2θ range spanning from 5° to 40° in 0.05° increments.

Functional group analysis was carried out via Fourier transform infrared (FT-IR) spectroscopy, employing a QATR-S model instrument, covering a frequency range of 500 to 4000 cm^−1^.

## 4. Conclusions

Various PLA/PBS composite fibers were successfully prepared using the melt-blown method with the addition of PBS into PLA, up to 10 wt%. The microstructure of the sheet samples indicated a high fiber density but a random arrangement of the fibers. SEM images and tensile tests showed that adding PBS to PLA enhanced the fiber size and surface smoothness, improving the alignment of the fibers and resulting in higher tensile strength with reduced elongation at the breakpoint. The addition of PBS significantly influenced the crystallinity, increasing it by approximately 1.46-fold, reaching 28.1% with a 10 wt% PBS addition. This enhancement in crystallinity lowered the T_g_ by about 1 °C and decreased the X_c_. In summary, when prepared through the melt-blown process, adding PBS to PLA, up to 10 wt%, enhanced the tensile strength, thermal properties, and crystallinity of the PLA/PBS composite compared to pure PLA. This novel nonwoven melt-blown fabric holds great potential for applications as a wound dressing. Future research should focus on exploring its properties and drug delivery capabilities. Plant fibers with a high surface area can improve mechanical and thermal qualities [38]. Due to its biocompatibility, it exhibits strong potential for use as a medical wound dressing in our future studies.

## Figures and Tables

**Figure 1 molecules-28-07215-f001:**
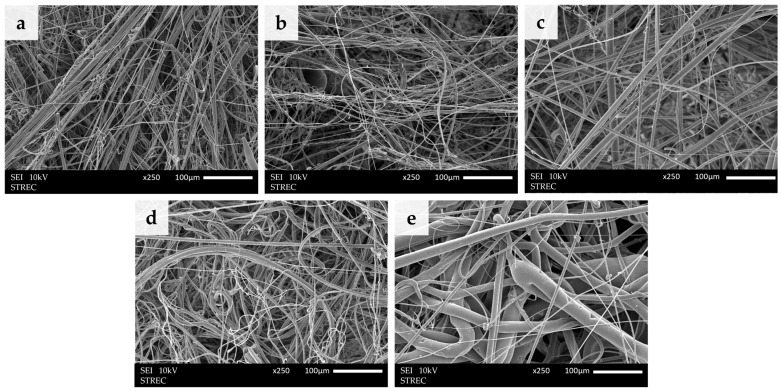
Scanning electron micrographs of the PLA/PBS blend fibers at various blend compositions (*w*/*w*) of (**a**) 100/0, (**b**) 97.5/2.5, (**c**) 95/5, (**d**) 90/10, and (**e**) 0/100 at 250× and the scale bar is 100 μm.

**Figure 2 molecules-28-07215-f002:**
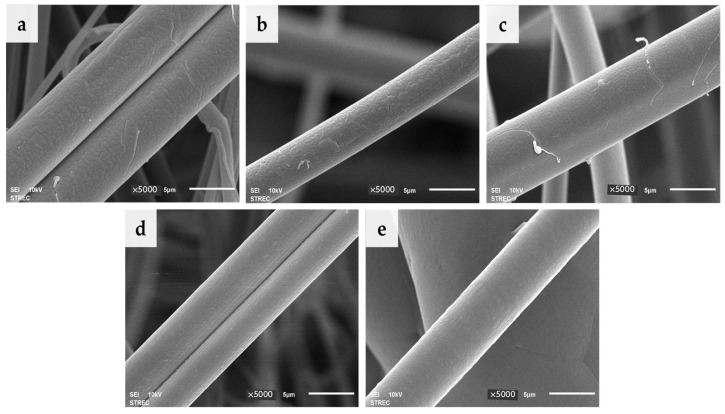
Scanning electron micrographs of PLA/PBS blend fibers with PLA/PBS blend compositions (*w*/*w*) of (**a**) 100/0, (**b**) 97.5/2.5, (**c**) 95/5, (**d**) 90/10, and (**e**) 0/100 at 5000×, and the scale bar is 5 μm.

**Figure 3 molecules-28-07215-f003:**
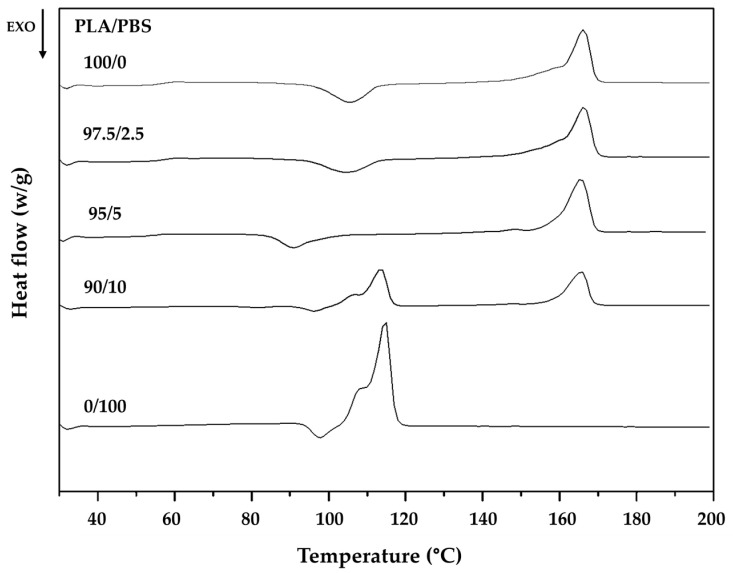
Representative heating DSC profiles of PLA/PBS fibers with different PLA/PBS blend ratios at a heating rate of 10 °C/min.

**Figure 4 molecules-28-07215-f004:**
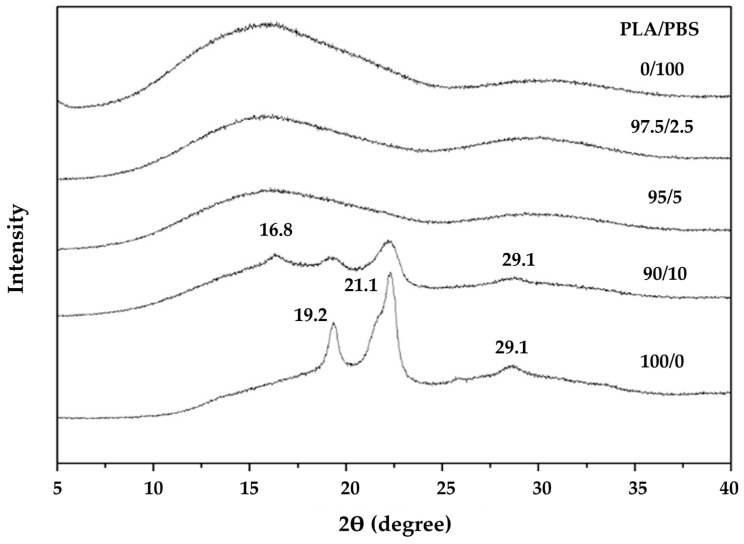
Representative XRD profiles of PLA/PBS composites with different PLA/PBS weight ratios.

**Figure 5 molecules-28-07215-f005:**
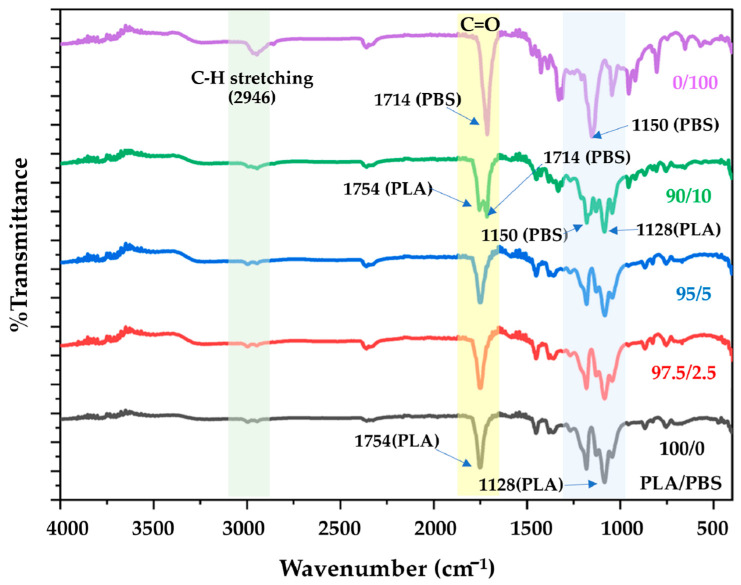
Representative FT-IR spectra of the PLA/PBS composites with different PLA/PBS weight ratios.

**Figure 6 molecules-28-07215-f006:**
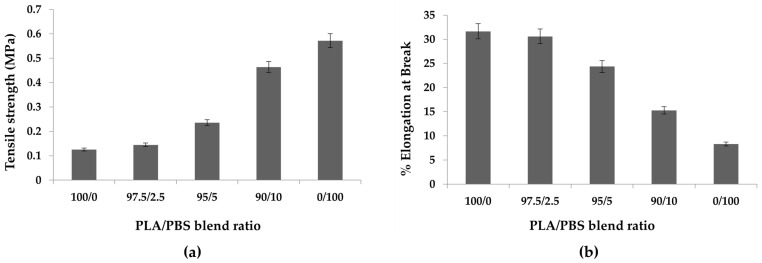
The (**a**) tensile strength and (**b**) elongation at break point percentage in PLA/PBS fiber composites with different PLA/PBS blends. Data are shown as the mean ± 1 standard deviation, derived from 9 repeats. Means with a different lowercase letter are significantly different (*p* < 0.05; one-way ANOVA).

**Figure 7 molecules-28-07215-f007:**
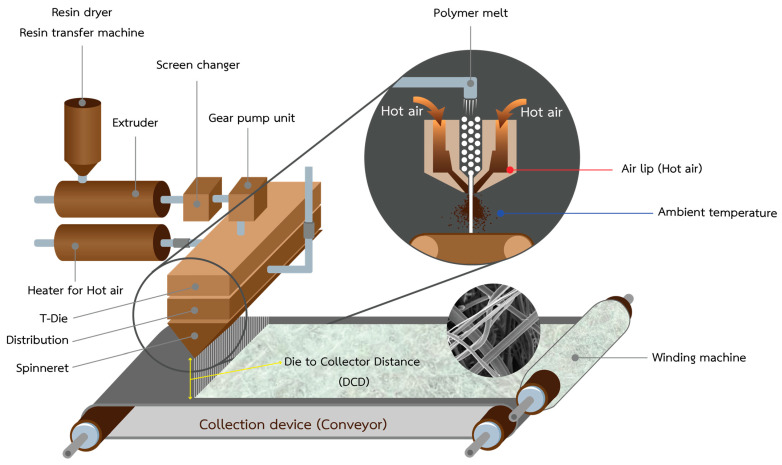
Schematic diagram showing the melt-blown process for nonwoven fiber production.

**Table 1 molecules-28-07215-t001:** Fabric formation ratios in the melt-blown process resulted in fabric-like structures’ formation (forming) or absence (non-forming).

PLA Concentration (%)	PBS Concentration (%)	Fabric Formation
100	0	Forming
97.5	2.5	Forming
95	5	Forming
90	10	Forming
80	20	Non-Forming
70	30	Non-Forming
60	40	Non-Forming
50	50	Non-Forming
0	100	Forming

**Table 2 molecules-28-07215-t002:** Thermal properties of the different PLA/PBS blends, as evaluated from the DSC analysis.

PLA/PBS Blend Ratio	PLA Phase					PBS Phase			
	T_g_ (°C)	T_m_ (°C)	T_cc_ (°C)	ΔH_m_ (J/g)	X_c_ (%)	T_m_ (°C)	T_cc_ (°C)	Δ	X_c_ (%)
100/0	60.7	166.0	105.8	51.2	19.2	-	-	-	-
97.5/2.5	60.5	166.3	104.5	51.8	20.4	-	-	-	-
95/5	60.2	166.3	96.2	50.4	23.9	-	-	-	-
90/10	59.2	165.8	91.7	52.5	28.1	113.5	96.2	48.5	40.7
0/100	-	-	-	-		114.6	97.8	80.4	64.8

**Table 3 molecules-28-07215-t003:** Melt-blowing process conditions for preparation of the nonwoven PLA/PBS fibers.

Parameter	Value
Temperature Extruder zone 1 (°C)	170
Temperature Extruder zone 2 (°C)	250
Temperature Extruder zone 3 (°C)	270
Temperature of die (°C)	275
Polymer flow rate (g/min)	13.2
Air pressure (MPa)	0.4
Hole diameter (mm)	0.35
Die to collector distance DCD (cm)	45

## Data Availability

Data available from the author on reasonable request.

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
