# Peer review of "Effect of Polybutylene Succinate Additive in Polylactic Acid Blend Fibers via a Melt-Blown Process"

_molecules, 2023, doi:10.3390/molecules28207215_

Round 1
Reviewer 1 Report
As a result of the evaluation of the publication, the following points came to the fore:
The introduction part needs a more straightforward presentation of the scientific gap. The aspects of the new study that differ from the results obtained in previous studies should be given more clearly here.
The presentation of raw data in the publication as a table or graphic is necessary; with them, the comparison is more straightforward. With 10% PBS incorporation, the TS value increased by approximately 400%, but the EatB value decreased by only 50%. The manuscript must include expressions that will bring this result to the fore. Perhaps the values expected from these mixtures and obtained values should be given on the graph explaining this significant difference, which is beyond the "rule of mixture."
It is written in line 253 that PBS does not increase the mechanical properties. They conducted only the tensile test, and the performance was good. It needs to be clarified why the authors wrote this sentence. This issue needs to be clarified.
Author Response
Reviewer #1 |
|
Comment |
Answer |
(1) The introduction part needs a more straightforward presentation of the scientific gap. The aspects of the new study that differ from the results obtained in previous studies should be given more clearly here. |
This has been revised accordingly. Line 39-45, 48-51 and 88-90.
|
(2) The presentation of raw data in the publication as a table or graphic is necessary; with them, the comparison is more straightforward. With 10% PBS incorporation, the TS (tensile strength) value increased by approximately 400%, but the EatB value decreased by only 50%. The manuscript must include expressions that will bring this result to the fore. Perhaps the values expected from these mixtures and obtained values should be given on the graph explaining this significant difference, which is beyond the "rule of mixture." |
We change section 3.2 to section 3.5. Therefore, we can continuously refer to crystallinity which affects to mechanical strength. Line 291-298.
When a material has high crystallinity, it typically means that its molecular chains are well-ordered, resulting in a stiff and brittle structure. This can indeed lead to high tensile strength but low elongation at break. PBS (Polybutylene Succinate) is a biodegradable thermoplastic with some flexibility. If you add a low concentration of PBS to another material, it may not act as a rubber-like component, especially if the other material has a high crystallinity and stiffness. Which is related to the DSC result.
The results are follow the "rule of mixture."
|
(3) It is written in line 253 that PBS does not increase the mechanical properties. They conducted only the tensile test, and the performance was good. It needs to be clarified why the authors wrote this sentence. This issue needs to be clarified. |
We already deleted this sentence because unclear of the information. |
Reviewer 2 Report
The manuscript entitled "Effect of Polybutylene succinate additive in Polylactic acid blend fibers via a melt-blown process". The Polybutylene succinate additive in Polylactic acid blend fibers were fabricated by a melt-blown process has been performed. The morphology, mechanical strength, thermal properties, crystallinity, chemical groups, and the water contact angle of as-prepared samples have been studied. Related comparative experiments were also performed. The experimental work is interesting. In my opinion, this work is interesting and has a certain reference in the development for the application in the fields of woven industry. However, there are some remarks that should be taken into consideration by the authors in order to raise this article to a good level for publication in Molecules.
The suggested modifications are listed as follows:
1. Many superscript problems need to be modified in detail.
2. In the introduction, the originality of the manuscript should be reformulated.
3. What are the advantages of the prepared composites compared with other literature?
4. The first letter of the horizontal coordinate in Figure 7 should be capitalized. The values of all characteristic peaks should be clearly marked on the graph.
5. The analysis of FTIR spectra should be based on relevant references.
Author Response
Reviewer #2 |
|
Comment |
Answer |
(1) Many superscript problems need to be modified in detail |
This has been revised accordingly. |
(2) In the introduction, the originality of the manuscript should be reformulated. |
This has been revised accordingly. |
(3) What are the advantages of the prepared composites compared with other literature? |
This has been revised accordingly. Line 48-51 |
(4) The first letter of the horizontal coordinate in Figure 7 should be capitalized. The values of all characteristic peaks should be clearly marked on the graph |
This has been revised accordingly. And Figure 7 changed to Figure 6. (Line 256) |
(5) The analysis of FTIR spectra should be based on relevant references. |
This has been revised accordingly. Line 261-282 |
Reviewer 3 Report
The manuscript titled “Effect of Polybutylene succinate additive in Polylactic acid blend fibers via a melt-blown process” by Tangnorawich, B.; et al. is a work where the authors study the performance of five different blends made of polylactic acid (PLA) and polybutylene succinate (PBS) [100/0, 97.5/2.5, 95/5, 90/10, 0/100]. For it, the authors employed several complementary techniques as scanning electron microscopy (SEM), tensile tests, differential scanning calorimetry (DSC), fourier-transform infrared spectroscopy (FTIR) and X-ray diffraction (XRD). The study is interesting and it is well-designed.
However, it exists some points that need to be addressed (please, see them below detailed point-by-point). The most relevant outcomes found by the authors can contribute in the growth of many fields like to design the next-generation of more durable composites for Industrial sectors as food packaging, aircraft, automotive, among others. For this reason, I will recommend the present scientific manuscript for further publication in Molecules once all the below described suggestions will be properly fixed.
Here, there exists some points that must be covered in order to improve the scientific quality of the manuscript paper:
1) ABSTRACT. “Differential scanning calorimetry” (line 28). Please, the authors should add the abbreviation of this term between brackets.
2) KEYWORDS (OPTIONAL). The authors should consider to add the assessed properties of PLA:PBS blends as terms in the keyword list (crystallinity, mechanical properties, surface morphology, …).
3) INTRODUCTION. The manuscript is in general tems poorly referenced (there are only 14 citations). The authors should do an effort in this regard. For example: “Examples of nonwoven processes include electrospinning, force spinning, melt spinning, spund bind, and melt blown” (lines 41-42). At least a relevant reference should be added for each mentioned technique.
4) “Among the above materials, PBS is the most used (…) for a rapid industrial molding process” (lines 58-60). Here, the authors should add a relevant reference [1].
[1] Aliotta, L.; et al. A Brief Review of Poly (Butylene Succinate) (PBS) and Its Main Copolymers: Synthesis, Blends, Composites, Biodegradability, and Applications. Polymers 2022, 14, 844. https://doi.org/10.3390/polym14040844.
5) EXPERIMENTAL. “Its melt flow indes (…) 24g/10 min, (…) and MFI of 27.69 g/10 min” (lines 87-90). Please, the authors should homogenize the significant figures of the shown data. This comment should be taken into account for the rest of the main manuscript body text.
6) “Samples were prepared at different PLA to PBS (…) 100/0, 97.5/2.5, 95/5, 90/10, and 0/100” (lines 105-106). Why did the authors use these PLA/PBS concentration ratios? A brief statement should be added in this regard.
7) “The mechanical properties (…) sheet (1 x 10 cm2)” (lines 126-127). “Then, the sample was cooled (…) under a N2 atmosphere” (lines 134-135). Please, the authors should introduce subscripts and superscripts where neccesary. Then, the authors need to add a reference where the mechanical properties of soft matter systems as blends are discussed [2].
[2] Magazzù, A.; et al. Investigation of Soft Matter Nanomechanics by Atomic Force Micrsocopy and Optical Tweezers: A Comprehensive Review. Nanomaterials 2023, 13, 693. https://doi.org/10.3390/nano13060963.
8) RESULTS AND DISCUSSION. Figure 2 caption “(a) polymer fractions formed on the fibers (magnification 250x)” (line 163). What is the condition that this magnification corresponds? This information should be detailed.
9) “3.4 Crystal structure characterization” (lines 211-223). Does the crystallization only affected by the PLA/PBS concentration ratio or also by the processing parameters [3]? Some information should be added in this regard.
[3] Su, S.; et al. Polylactide (PLA) and Its Blends with Poly(butylene succinate) PBS): A Brief Review. Polymers 2019, 11, 1193. https://doi.org/10.3390/polym11071193.
10) Figure 6 (OPTIONAL, line 224). The authors should consider to modify “theta” by its respective greek letter in the X-axis lettering.
11) Figure 7 (line 227). Please, the authors should modify “wavelength” by “wavenumber”.
12) Then, the authors only considered the wavenumber associated to the carbonyl group. Why did they not take into account the band attributed to the stretching vibration of –CH3 in folded PLA chains and the bending vibration of –OH groups of PBS (nearby 1130 cm-1 amd 1160 cm-1, respectively). These wavenumbers should slightly decrease when the different blends of PLA/PBS are formed. What is the opinion of the authors? Some information should be added in this regard.
13) CONCLUSIONS. The most relevant outcomes found in this work are perfectly remarked. The authors need to discuss about potential future perspectives to strengthen the importance of this research. In this context, it is relevant to show the synergistic effect to use the blends of PLA/PBS with plant fibers [4] or carbon particles [5] to enhance their physico-chemical properties.
[4] Marcuello, C.; et al. Influence of Surface Chemistry of Fiber and Lignocellulosic Materials on Adhesion Properties with Polybutylene Succinate at Nanoscale. Materials 2023, 16, 2440. https://doi.org/10.3390/ma16062440.
[5] Wang, X.; et al. Study of carbon black-filled poly(butylene succinate)/polylactide blend. J. Appl. Polym. Sci. 2012, 126, 1876-1884. https://doi.org/10.1002/app.36944.
14) PATENTS (line 254). Please, the authors should change the name of this section by “funding”.
15) REFERENCES. The references are not in the proper format style of Molecules. The journal name should appear in Italics and abbreviated form, the publication year highlighted in bold, among other minor corrections. The authors should take care of this point.
The authors should recheck the English out before to send the manuscript revised version in order to fix some existing typos.
Author Response
Reviewer #3 |
|
Comment |
Answer |
(1) ABSTRACT. “Differential scanning calorimetry” (line 28). Please, the authors should add the abbreviation of this term between brackets |
This has been revised accordingly. |
(2) KEYWORDS (OPTIONAL). The authors should consider to add the assessed properties of PLA:PBS blends as terms in the keyword list (crystallinity, mechanical properties, surface morphology, …). |
This has been revised accordingly. |
(3) INTRODUCTION. The manuscript is in general tems poorly referenced (there are only 14 citations). The authors should do an effort in this regard. For example: “Examples of nonwoven processes include electrospinning, force spinning, melt spinning, spund bind, and melt blown” (lines 41-42). At least a relevant reference should be added for each mentioned technique. |
This has been revised accordingly. Reference [3] |
(4) “Among the above materials, PBS is the most used (…) for a rapid industrial molding process” (lines 58-60). Here, the authors should add a relevant reference [1] [1] Aliotta, L.; et al. A Brief Review of Poly (Butylene Succinate) (PBS) and Its Main Copolymers: Synthesis, Blends, Composites,Biodegradability, and Applications. Polymers 2022, 14, 844. https://doi.org/10.3390/polym14040844. |
This has been revised accordingly. Reference [7] |
(5) EXPERIMENTAL. “Its melt flow indes (…) 24g/10 min, (…) and MFI of 27.69 g/10 min” (lines 87-90). Please, the authors should homogenize the significant figures of the shown data. This comment should be taken into account for the rest of the main manuscript body text |
PLA and PBS were mixed with high-speed homogenizer two components until homogenized then go to Melt blown process |
(6) Samples were prepared at different PLA to PBS (…) 100/0, 97.5/2.5, 95/5, 90/10, and 0/100” (lines 105-106). Why did the authors use these PLA/PBS concentration ratios? A brief statement should be added in this regard. |
Table 2. Fabric formation ratios in the melt-blown process resulted in the formation (forming) or absence (non-forming) of fabric-like structures. Line 169-171 and 189-190 and 121-124 (Reference 10-15) |
(7) “The mechanical properties (…) sheet (1 x 10 cm2)” (lines 126-127). “Then, the sample was cooled (…) under a N2 atmosphere” (lines 134-135). Please, the authors should introduce subscripts and superscripts where neccesary. Then, the authors need to add a reference where the mechanical properties of soft matter systems as blends are discussed [2]. [2] Magazzù, A.; et al. Investigation of Soft Matter Nanomechanics by Atomic Force Micrsocopy and Optical Tweezers: A Comprehensive Review. Nanomaterials 2023, 13, 693. https://doi.org/10.3390/nano13060963. |
This has been revised accordingly. Reference [13] |
(8) RESULTS AND DISCUSSION. Figure 2 caption “(a) polymer fractions formed on the fibers (magnification 250x)” (line 163). What is the condition that this magnification corresponds? This information should be detailed |
Figure 2a has been cut and has been revised accordingly. Line 182-186
|
(9) “3.4 Crystal structure characterization” (lines 211-223). Does the crystallization only affected by the PLA/PBS concentration ratio or also by the processing parameters [3]? Some information should be added in this regard. [3] Su, S.; et al. Polylactide (PLA) and Its Blends with Poly(butylene succinate) PBS): A Brief Review. Polymers 2019,11, 1193. https://doi.org/10.3390/polym11071193. |
In polymer blends, when one component is initially at a low concentration, the dispersed particles can have either spherical or fibrillar shapes, forming a dispersed phase-matrix morphology. However, as the concentration of the minor phase increases, the particles start to come closer and eventually reach a percolation threshold. Beyond this point, more of the minor component is integrated into a single percolating structure, resulting in dual-phase continuity or co-continuity. In this morphological structure, both phases remain continuously connected throughout the bulk of the blend. |
(10) Figure 6 (OPTIONAL, line 224). The authors should consider to modify “theta” by its respective greek letter in the X-axis lettering. |
This has been revised accordingly.
|
(11) Figure 7 (line 227). Please, the authors should modify “wavelength” by “wavenumber”. |
This has been revised accordingly. |
(12) Then, the authors only considered the wavenumber associated to the carbonyl group. Why did they not take into account the band attributed to the stretching vibration of –CH in folded PLA chains and the bending vibration of –OH groups of PBS (nearby 1130 cm and 1160 cm, respectively). These wavenumbers should slightly decrease when the different blends of PLA/PBS are formed. What is the opinion of the authors? Some information should be added in this regard. |
This has been added accordingly. |
(13) CONCLUSIONS. The most relevant outcomes found in this work are perfectly remarked. The authors need to discuss about potential future perspectives to strengthen the importance of this research. In this context, it is relevant to show the synergistic effect to use the blends of PLA/PBS with plant fibers [4] or carbon particles [5] to enhance their physico-chemical properties. [4] Marcuello, C.; et al. Influence of Surface Chemistry of Fiber and Lignocellulosic Materials on Adhesion Properties with Polybutylene Succinate at Nanoscale. Materials 2023, 16, 2440. https://doi.org/10.3390/ma16062440. [5] Wang, X.; et al. Study of carbon black-filled poly(butylene succinate)/polylactide blend. J. Appl. Polym. Sci. 2012, 126, 1876-1884. https://doi.org/10.1002/app.36944. |
The conclusion has been revised accordingly. And also add into Reference [24] Plant fiber high surface area, able to increase mechanical and thermal properties. Its have biocompatibility then high potential to use as the medical wound dressing as our future work |
(14) PATENTS (line 254). Please, the authors should change the name of this section by “funding”. |
This has been revised accordingly. |
(15) REFERENCES. The references are not in the proper format style of Molecules. The journal name should appear in Italics and abbreviated form, the publication year highlighted in bold, among other minor corrections. The authors should take care of this point. |
This has been revised accordingly. |
Reviewer 4 Report
The paper “molecules-2573564” is devoted to study of effect of PBS concentration on morphology, thermal, mechanical and crystalline properties of the PLA-based nonwoven materials. Although it describes new results, in my opinion it lacks fundamental novelty in terms of chemistry. It would be advantageous to present not only factual information regarding how PBS concentration affects the properties of the material, but also speculation on why it affects them the way it does. For that it may be suitable to perform hot-stage optical microscopy of the mixtures for observation of the crystallization process. Neverthless, the paper may be published in the “Molecules” journal after a major revision with addition of optical microscopy data, some speculation regarding mechanism of PBS effect on the properties of the material and accounting for some specific comments below.
1. It seems that some references are missing in the introduction section. For example in Lines 53-56 the authors list some polymers that can be blended with PLA. However only one reference is given. Concrete references for research articles where addition of these polymers were evaluated would be appreciated.
2. Lines 70,71. How do the plasticization affects acid-base resistance?
3. Line 85. Please add information regarding the PLA used. Was it all-left (PLLA), all-right or enantiomer mixture?
4. Lines 86,87. How did the authors define melting temperature? In fact in most semicrystalline polymer melting process begins right after glass transition and continues until equilibrium melting temperature. So the range of melting process is much wider. However, for technical purposes it is generally enough to define peak temperature in DSC heating curve or temperature at which the polymer acquires ability to flow.
5. Lines 87, 90. Please describe experimental conditions for melt flow index measurements (temperature, load, capillary diameter etc.)
6. Line 99. In text it is stated that “hot air” is used while in Figure 1 “cooling air” is shown. Which is correct? Or maybe air being hot still acts as a coolant for even more hot fibers?
7. Table 1. Please explain why polymer flow rate of 13.2 g/min, temperature of die of 275°C and other parameters of the process were chosen.
8. Line 129. It seems that electrospinning was not used in the paper. In this case what is the direction the authors speak about?
9. Line 132,133,135. “2” should be subscript.
10. Section 2.3. Please add which exact instruments were used for DSC, SEM, XRD and FTIR investigation.
11. Line 148. Why precursor feeding was intermittent? Is it possible to make it continuous?
12. Line 152-160. Since the fibers were coated with gold for SEM study, can it be claimed that the smoothness is a characteristic of fibers, not golden coating?
13. Line 172. What is the state of different PLA/PBS mixtures at room temperature? Is it phase separated (into PLA-rich and PBS-rich domains)? If yes, how can the authors judge on evenness of PBS distribution?
14. Line 177. Is it mechanical strength of fibers or sheets? If the properties described are those of sheets, how high was the porosity of the sheets and was it dependent on the PLA/PBS ratio?
15. Line 199. “Crystallinity degree” is probably miswritten as “Crystallization rate”. Also, please describe in experimental part how crystallinity degree was calculated.
16. Sentence in Line 202-204 is weird. Please reformulate to make it more readable.
17. Figure 5 caption. It should be noted that the thermograms presented are obtained during second heating (if correct).
18. Lines 219-222. XRD results indicate that there could be some crystallinity in samples right after their formation. Is it confirmed by appearance of PLA melting peak on DSC-thermograms of the samples obtained in first heating?
19. Section 3.5. What was the purpose of FTIR analysis? In connection to question 13, is it possible to speculate on state of the mixtures (Is it phase separated or in amorphous regions the mixture is homogeneous)?
There are some misprints and grammar mistakes in the paper text. Please revise carefully.
Author Response
Reviewer #4 |
|
Comment |
Answer |
(1) It seems that some references are missing in the introduction section. For example in Lines 53-56 the authors list some polymers that can be blended with PLA. However only one reference is given. Concrete references for research articles where addition of these polymers were evaluated would be appreciated. |
This has been revised accordingly. Ref [5] |
(2) How do the plasticization affects acid-base resistance? |
Suggestion to use at neutral pH because this work is applied in medical wound dressing |
(3) Line 85. Please add information regarding the PLA used. Was it all-left (PLLA), all-right or enantiomer mixture? |
This has been revised accordingly. |
(4) Lines 86,87. How did the authors define melting temperature? In fact in most semicrystalline polymer melting process begins right after glass transition and continues until equilibrium melting temperature. So the range of melting process is much wider. However, for technical purposes it is generally enough to define peak temperature in DSC heating curve or temperature at which the polymer acquires ability to flow |
This has been revised accordingly. |
(5) Lines 87, 90. Please describe experimental conditions for melt flow index measurements (temperature, load, capillary diameter etc.) |
This has been revised accordingly. Temperature at 180 – 250 °C, load 2.16 kg |
(6) . Line 99. In text it is stated that “hot air” is used while in Figure 1 “cooling air” is shown. Which is correct? Or maybe air being hot still acts as a coolant for even more hot fibers? |
This has been revised accordingly. Change from cooling air to ambient temperature in Figure1 |
(7) Table 1. Please explain why polymer flow rate of 13.2 g/min, temperature of die of 275°C and other parameters of the process were chosen. |
This has been revised accordingly. According to our preliminary experiment to find out good experimental conditions the result is a processing parameter which is presented in Table 1. |
(8) Line 129. It seems that electrospinning was not used in the paper. In this case what is the direction the authors speak about? |
This has been revised accordingly.
|
(9) Line 132,133,135. “2” should be subscript. |
This has been revised accordingly.
|
(10) Section 2.3. Please add which exact instruments were used for DSC, SEM, XRD and FTIR investigation. |
This has been revised accordingly. In section Section 2.3 |
(11) Line 148. Why precursor feeding was intermittent? Is it possible to make it continuous? |
It’s the nature of the high-pressure process. The non-continually feeding does not affect the properties of materials. |
(12) Line 152-160. Since the fibers were coated with gold for SEM study, can it be claimed that the smoothness is a characteristic of fibers, not golden coating? |
Gold sputter coating, or gold metallization, is a common practice in scanning electron microscopy (SEM). The process was carefully control the thickness of the gold coating. It has been controlled to prevent overcoating, which can obscure fine surface details. |
(13) Line 172. What is the state of different PLA/PBS mixtures at room temperature? Is it phase separated (into PLA-rich and PBS-rich domains)? If yes, how can the authors judge on evenness of PBS distribution? |
Because PBS was added in low concentration in PLA is can not accumulate but fully dispersion in the PLA matrix |
(14) Line 177. Is it mechanical strength of fibers or sheets? If the properties described are those of sheets, how high was the porosity of the sheets and was it dependent on the PLA/PBS ratio? |
It is the mechanical strength of sheets. The random arrangement of microfibers results in a nonwoven fabric with unique properties such as high surface area and porosity. But in our work, we did not study porosity. It may be considered in the future after we decide to load the drug or other application |
(15) Line 199. “Crystallinity degree” is probably miswritten as “Crystallization rate”. Also, please describe in experimental part how crystallinity degree was calculated. |
This has been revised and added the equation accordingly.
|
(16) Sentence in Line 202-204 is weird. Please reformulate to make it more readable. |
The reduction in the Tcc can be attributed to the PBS formation that led to PLA fiber growth, as seen in the increasing Xc with increasing PBS content. Thus, adding PBS to PLA can enhance the thermal resistivity and crystallinity of blends [12].
|
(17) Figure 5 caption. It should be noted that the thermograms presented are obtained during second heating (if correct). |
This result of DSC analysis is a versatile technique that measures heat flow during changes in temperature, and it already test both "endo" (endothermic) and "exo" (exothermic) processes. |
(18) Lines 219-222. XRD results indicate that there could be some crystallinity in samples right after their formation. Is it confirmed by appearance of PLA melting peak on DSC thermograms of the samples obtained in first heating? |
The answer is Yes We can see it in the DSC result |
(19) Section 3.5. What was the purpose of FTIR analysis? In connection to question 13, is it possible to speculate on state of the mixtures (Is it phase separated or in amorphous regions the mixture is homogeneous)? |
This has been revised and accordingly. Line 262-265
|
Reviewer 5 Report
The manuscript's topic is interesting and within the journal's scope.
Overall, the Abstract is well prepared. Here, I would recommend that the respected authors swap the first (lines 23-24) and second (lines 24-26) sentences and add an introductory sentence about the relevance of the research.
In the Introduction: Lines 63-64, "In the past, there have been many studies to improve the properties of PLA through various preparation methods, but these have yielded different results." This statement needs references. Please add those.
I ask the authors to justify further the experimental setup: "In this study, PLA/PBS was prepared by a melt-blown process at different PLA: PBS (w/w) ratios" (lines 76-77). The current rationale is weak. That is, I am asking for additional references from previous studies and, in that way, to outline the need for the presented research.
In the Materials and methods - it is not clear why such weight ratios were used - "Samples were prepared at different PLA to PBS weight ratios of 100/0, 97.5/2.5, 95/5, 90/10, and 0/100 at various temperatures from 180–250 °C for 10 seconds". That is, I ask that the experimental plan and the corresponding values of the regime factors be justified, if possible, based on previous studies (references).
In the results and discussion – in my opinion, the data analysis is high, but the comparative analysis with previous studies is relatively weak. That is precisely my recommendation to the respected authors, namely, to add additional references and significantly increase (improve) the comparative analysis.
The Conclusions reflect the main results of the research, but its necessity and, above all, the main novelty of the work are not highlighted.
The references cited are appropriate but, in my opinion, insufficient.
Overall, the research was carried out to a high standard, but the manuscript needs some improvements before its acceptance for publication.
Author Response
Reviewer #5 |
|
Comment |
Answer |
(1) the respected authors swap the first (lines 23-24) and second (lines 24-26) sentences and add an introductory sentence about the relevance of the research |
This has been improved and revised accordingly. Line 23-26
|
(2) In the Introduction: Lines 63-64, "In the past, there have been many studies to improve the properties of PLA through various preparation methods, but these have yielded different results." This statement needs references. Please add those |
This has been improved and revised accordingly. Line 48-52.
|
(3) I ask the authors to justify further the experimental setup: "In this study,PLA/PBS was prepared by a melt-blown process at different PLA: PBS (w/w) ratios" (lines 76-77). The current rationale is weak. That is, I am asking for additional references from previous studies and, in that way, to outline the need for the presented research |
This has been improved and revised accordingly.
|
(4) In the Materials and methods - it is not clear why such weight ratios were used - "Samples were prepared at different PLA to PBS weight ratios of 100/0, 97.5/2.5, 95/5, 90/10, and 0/100 at various temperatures from 180–250 °C for 10 seconds". That is, I ask that the experimental plan and the corresponding values of the regime factors be justified, if possible, based on previous studies (references). |
This has been improved and revised accordingly. Line 169-171 and 189-190 (Table2)
|
(5) In the results and discussion – in my opinion, the data analysis is high, but the comparative analysis with previous studies is relatively weak. That is precisely my recommendation to the respected authors, namely, to add additional references and significantly increase (improve) the comparative analysis. |
This has been improved and revised accordingly.
|
(6) The Conclusions reflect the main results of the research, but its necessity and, above all, the main novelty of the work are not highlighted. |
This has been improved and revised accordingly.
|
Round 2
Reviewer 2 Report
I think the revised version can be accepted for publication.
Author Response
Thank you for your consideration and suggestion
Reviewer 3 Report
The authors did a great deal of effort and for this reason the scientific quality of the manuscript was greatly improved. Based on the relevance of the most important outcomes found by the authors and the scope of the journal I may recommend this work for further publication in Molecules.
As final minor remark the authors should fix the following sentence:
"The researcher team need to study" by "The research team needs (...)".
The paper is well-written. The authors should do a final check just in case.
Author Response
Revised accordingly line 326 and Thank you for your consideration and suggestion
Reviewer 4 Report
First of all I should note that it was difficult to review the revised version of the manuscript № molecules-2573564 since it was hard to keep track of the changes. Some portions of the text are marked yellow, the other are added using “track changes” function. Also, most of the answers given by the authors are just “this has been revised accordingly”. It is advised not only to confirm that the comment was taken into account but also to show what exact portions of the paper text were revised (line numbers). Also, with such manner of responses some questions remained unanswered. It is ok to state that something was revised according to the comment when the comment is just a proposal for concrete changes. However when the comment represents a question to the authors, the response should contain direct answer.
The paper needs another round of major revision before it can be accepted for publication. At this point, I think even the authors got confused with the amount of changes made to the paper text. Thus, another "clean" version of the manuscript needs to be reviewed. Also, hot stage microscopy in polarized light data is needed to support some claims made by the authors. The specific comments that need to be accounted for are the following.
1. In my opinion the introduction section is still missing some important background references. For example, the revised sentence in Line 57 is “Examples of nonwoven processes include electrospinning, force spinning, melt spinning, spun bind, and melt blown. [5].” seems to give reference only for the electrospinning process, while references for the other methods are still absent. Also, the paragraph in Lines 67-78 is still referenced poorly. It should also be noted that some references are not cited in the main text, for example [23].
2. Question 2 from the previous round of revision remained unaccounted for. In lines 92-94 of the revised version it is stated that “…PBS displayed a plasticizer effect, which enchanced… …acid-base resistance…”. Do the authors imply that placticization is a reason for increased acid-base resistance? If yes, what is the mechanism of plasticizer effect on the chemical resistance of the polymer? If no (probably it is the case), the sentence must be revised.
3. Although the authors replied to the question 3 of the previous review with standard answer “this has been revised accordingly”, in fact this information is still missing in the paper text. Even though I can read between the lines that all-L enantiomer (PLLA) was used, it should be noted explicitly.
4. Line 113. Again, how melting point of 160–170°C was determined? In my opinion it would be better to specify peak maximum temperature from DSC data (if possible, with experimental error) instead of abstract range.
5. The authors contradict themselves in Line 229 and response to the question 13 of the previous review. In the paper text it is stated that “…PBS phase was evenly distributed…”. At the same time in the response the authors wrote “Because PBS was added in low concentration in PLA is cannot accumulate but fully dispersion in the PLA matrix”. If there is a PHASE of PBS, according to term “phase” definition, it cannot be fully dispersed in the PLA matrix. Moreover, DSC data shows that for the 90/10 mixture both components can crystallize. For crystallization of both components some phase separation must occur. Therefore polarized light optical microscopy data (hot stage POM) is needed to assess phase state of different mixtures. In my opinion the paper should not be published in “molecules” journal without addition of such data.
6. Term “phase” is incorrectly used in Lines 290-297. And again, the authors contradict themselves writing that “in polymer blends… the dispersed particles can have either spherical or fibrillar shapes…”. If there are “particles” in the mixture, this mixture is phase separated, not uniform. Uniformity of the particles dispersion throughout the matrix is another question.
8. In lines 312-313 it is stated that “FT-IR can provide insights into the interactions between PLA and PBS molecules”. However the authors did not make any specific conclusions on this matter from the FTIR data.
7. In reply to the comment 14 of the previous review the authors noted that they did not study the porosity of the nonwoven sheets. In my opinion data on tensile strength of nonwoven sheets is not meaningful without corresponding data on their porosity. The authors linked the increase of tensile strength of the sheets with decrease of PLLA crystallinity degree in the samples. However if different sheets have different porosity, such behavior can be explained by decrease of porosity of the sheets with addition of PBS.
Minor spellcheck will be required.
Author Response
Reviewer #4 |
|
Comment |
Answer |
1. In my opinion the introduction section is still missing some important background references. For example, the revised sentence in Line 57 is “Examples of nonwoven processes include electrospinning, force spinning, melt spinning, spun bind, and melt blown. [5].” seems to give reference only for the electrospinning process, while references for the other methods are still absent. Also, the paragraph in Lines 67-78 is still referenced poorly. It should also be noted that some references are not cited in the main text, for example [23]. |
We already added references for each method. From the sentence, “Examples of nonwoven processes include electrospinning, force spinning, melt spinning, spun bind, and melt blown. References number [5-9].
Lines 67-78 are already added referenced in [12-15].
Reference [23] now changed to [31] and also changed to Moreira, A. C. F., et al. (2003). "Co-continuous morphologies in polystyrene/ethylene–vinyl acetate blends: The influence of the processing temperature." Journal of Applied Polymer Science 89(2): 386-398. |
2. Question 2 from the previous round of revision remained unaccounted for. In lines 92-94 of the revised version, it is stated that “…PBS displayed a plasticizer effect, which enchanced… …acid-base resistance…”. Do the authors imply that placticization is a reason for increased acid-base resistance? If yes, what is the mechanism of plasticizer effect on the chemical resistance of the polymer? If no (probably it is the case), the sentence must be revised. |
We already edited it as “Moreover, PBS displayed a plasticizer effect, which enhanced the flexibility, elasticity, and thermal resistance properties of the obtained PLA/PBS composite fibers compared to the pure PLA fibers [20]” Line 83-85 |
3. Although the authors replied to the question 3 of the previous review with standard answer “this has been revised accordingly”, in fact this information is still missing in the paper text. Even though I can read between the lines that all-L enantiomer (PLLA) was used, it should be noted explicitly. |
Thank you for your suggestion, following the previous review, we can not specify whether is it an all-left (PLLA), all-right, or enantiomer mixture. Following the MSDS (The PLA (NatureWorks® Ingeo™ 3251D Injection Grade PLA) also does not show this information. (Line 100) |
4. Line 113. Again, how melting point of 160–170°C was determined? In my opinion it would be better to specify peak maximum temperature from DSC data (if possible, with experimental error) instead of abstract range. |
Revised in Line102 as “melting point (Tm) 170°C, Glass Transition Temp (Tg) 65.0°C” |
5. The authors contradict themselves in Line 229 and response to the question 13 of the previous review. In the paper text it is stated that “…PBS phase was evenly distributed…”. At the same time in the response the authors wrote “Because PBS was added in low concentration in PLA is cannot accumulate but fully dispersion in the PLA matrix”. If there is a PHASE of PBS, according to term “phase” definition, it cannot be fully dispersed in the PLA matrix. Moreover, DSC data shows that for the 90/10 mixture both components can crystallize. For crystallization of both components some phase separation must occur. Therefore polarized light optical microscopy data (hot stage POM) is needed to assess phase state of different mixtures. In my opinion the paper should not be published in “molecules” journal without addition of such data. |
Revised in lines 234-240 |
6. Term “phase” is incorrectly used in Lines 290-297. And again, the authors contradict themselves writing that “in polymer blends… the dispersed particles can have either spherical or fibrillar shapes…”. If there are “particles” in the mixture, this mixture is phase separated, not uniform. Uniformity of the particles dispersion throughout the matrix is another question. |
Revised in Lind 258-260 as “In polymer blends, when one component is initially at a low concentration, the dispersed particles can have mainly fibrillar shapes, forming a dispersed phase-matrix morphology” and for more detail show in result and discussion of XRD and DSC |
7. In reply to the comment 14 of the previous review the authors noted that they did not study the porosity of the nonwoven sheets. In my opinion data on tensile strength of nonwoven sheets is not meaningful without corresponding data on their porosity. The authors linked the increase of tensile strength of the sheets with decrease of PLLA crystallinity degree in the samples. However if different sheets have different porosity, such behavior can be explained by decrease of porosity of the sheets with addition of PBS. |
Revised in lines 319-336 |
8. In lines 312-313 it is stated that “FT-IR can provide insights into the interactions between PLA and PBS molecules”. However the authors did not make any specific conclusions on this matter from the FTIR data. |
We added the explanation in Line 279-300. “It shows the position of the peak, peak widths at half height and percentage integral absorbance of the component of C=O bands are similar as the PLA/PBS blends.” And “The shifting bonds caused by strengthening of bond energy of backbones and side groups were a consequence of strong inter/ intra-molecular interactions. In addition to formation of hydrogen bonds between the oxygen functional groups carried by PLA and PBS.” |
Reviewer 5 Report
The manuscript is significantly improved as the esteemed authors have complied or taken a stance on all my recommendations. That gives me a reason to recommend acceptance of the manuscript in its present form.
Author Response

(The authors gave the same response as above.)

Round 3
Reviewer 4 Report
Despite I personally am disappointed that the authors decided not to accept two of my major suggestions regarding porosity measurement and investigation of phase equilibria/thermal behavior of the PLA/PBS mixtures, which, in my opinion would improve the paper, the revised version of the manuscript can be published in "molecules journal".
Author Response
Thank you for taking the time to review our research paper and for your insightful comments. We appreciate your effort and the valuable suggestions you provided.
However, we regret to inform you that we cannot implement the corrections you recommended due to the nature of this research project. The study is conducted in collaboration with a private company, which has imposed restrictions on the information that can be disclosed publicly. As a result, we are limited in the amount of data and details we can include in the published research.
Please understand that this decision does not reflect our unwillingness to consider your recommendations; rather, it is a constraint imposed by the confidentiality requirements of the private entity involved. We hope you understand the context of this limitation and appreciate your understanding in this matter.
Once again, we sincerely thank you for your time and effort in providing feedback on our research. Your engagement is highly valued, and we look forward to any future opportunities to collaborate or receive your input.